# LEARNED OPTIMIZERS THAT OUTPERFORM ON WALL-CLOCK AND VALIDATION LOSS

## ABSTRACT

Deep learning has shown that learned functions can dramatically outperform hand-designed functions on perceptual tasks. Analogously, this suggests that learned optimizers may similarly outperform current hand-designed optimizers, especially for specific problems. However, learned optimizers are notoriously difficult to train and have yet to demonstrate wall-clock speedups over hand-designed optimizers, and thus are rarely used in practice. Typically, learned optimizers are trained by truncated backpropagation through an unrolled optimization process. The resulting gradients are either strongly biased (for short truncations) or have exploding norm (for long truncations). In this work we propose a training scheme which overcomes both of these difficulties, by dynamically weighting two unbiased gradient estimators for a variational loss on optimizer performance. This allows us to train neural networks to perform optimization faster than well tuned first-order methods. Moreover, by training the optimizer against validation loss (as opposed to training loss), we are able to learn optimizers that train networks to better generalization than first order methods. We demonstrate these results on problems where our learned optimizer trains convolutional networks in a fifth of the wall-clock time compared to tuned first-order methods, and with an improvement in validation loss.

## 1 INTRODUCTION

Gradient based optimization is a cornerstone of modern machine learning. Improvements in optimization have been critical to recent successes on a wide variety of problems. In practice, this typically involves analysis and development of hand-designed optimization algorithms (Nesterov, 1983; Duchi et al., 2011; Tieleman & Hinton, 2012; Kingma & Ba, 2014). These algorithms generally work well on a wide variety of tasks, and are tuned to specific problems via hyperparameter search. On the other hand, a complementary approach is to *learn* the optimization algorithm (Bengio et al., 1990; Schmidhuber, 1995; Hochreiter et al., 2001; Andrychowicz et al., 2016; Wichrowska et al., 2017; Li & Malik, 2017b; Bello et al., 2017). That is, to learn a function to perform optimization, targeted to particular problems of interest. In this way, the algorithm may learn task specific structure, enabling dramatic performance improvements over more general optimizers.

However, training learned optimizers is notoriously difficult. Existing work in this vein can be classified into two broad categories. On one hand are black-box methods such as evolution (Goldberg & Holland, 1988; Bengio et al., 1992), random search (Bergstra & Bengio, 2012), reinforcement learning (Bello et al., 2017; Li & Malik, 2017a;b), or Bayesian optimization (Snoek et al., 2012). However, these methods scale poorly with the number of optimizer parameters. The other approach is to use first-order methods, by computing the gradient of some measure of optimizer effectiveness with respect to the optimizer parameters. Computing these gradients is costly as we need to iteratively apply the learned update rule, and then backpropagate through these applications, a technique commonly referred to as "unrolled optimization" (Bengio, 2000; Maclaurin et al., 2015). To address the problem of backpropagation through many optimization steps (analogous to many timesteps in recurrent neural networks), many works make use of truncated backpropagation though time (TBPTT) to partition the long unrolled computational graph into separate pieces (Werbos, 1990; Tallec & Ollivier, 2017). This yields computational savings, but at the cost of increased bias (Tallec & Ollivier, 2017) and/or exploding gradients due to many iterated update steps (Pascanu et al., 2013; Parmas et al., 2018). Existing methods address the bias at the cost of increased variance or computa-

tional complexity (Williams & Zipser, 1989; Ollivier et al., 2015; Tallec & Ollivier, 2017). Previous techniques for training RNNs via TBPTT have thus far not been effective for training optimizers.

In this paper, we analytically and experimentally explore the debilitating role of bias and exploding gradients on training optimizers (§2.3). We then show how these pathologies can be remedied by optimizing the parameters of a distribution over the optimizer parameters, known as variational optimization (Staines & Barber, 2012) (§3). We define two unbiased gradient estimators for this objective: a reparameterization based gradient, and evolutionary strategies (Rechenberg, 1973; Nesterov & Spokoiny, 2011). By dynamically reweighting the contribution of these two gradient estimators, we are able to avoid strongly biased or exploding gradients, and thus stably and efficiently train learned optimizers.

We demonstrate the utility of this approach by training a learned optimizer to target optimization of convolutional networks on image classification (§4). On the targeted task distribution, this learned optimizer achieves better validation loss, and is five times faster in *wall-clock time*, compared to well tuned hand-designed optimizers such as SGD+Momentum, RMSProp, and ADAM (Figure 1). While not the explicit focus of this work, we also find that the learned optimizer demonstrates promising generalization ability on out of distribution tasks (Figure 6).

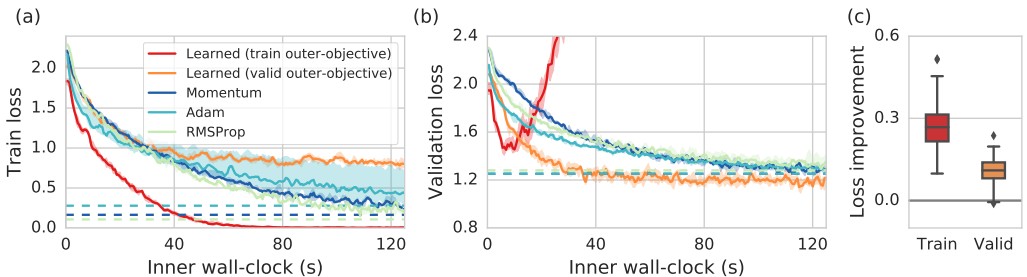

Figure 1: Learned optimizers outperform existing optimizers on training loss (a) and validation loss (b). **(a,b)** Training and validation curves for a three layer CNN, trained on a subset of 32x32 imagenet classes not seen during outer-training of the optimizer. Dashed lines indicate the best achieved performance over an additional 130 seconds. We show two learned optimizers – one trained to minimize training loss, and the other trained to minimize validation loss on the inner-problem. We compare against Adam, RMSProp, and SGD+Momentum, individually tuned for the train and validation loss (Panel (a) and (b), respectively). On training loss (a), our learned optimizer approaches zero training loss, and achieves it's smallest loss values in less than one quarter the wall-clock time. On validation loss (b), our learned optimizer achieves a lower minimum, in roughly one third the wall-clock time. Shaded regions correspond to 25 and 75 percentile over five random initializations of the CNN. For plots showing performance in terms of step count rather than wall clock (where we achieve even more dramatic speedups), and for more task instances, see Appendix D. **(c)** Distribution of the performance difference between the learned optimizers and a tuned baseline of either Adam, RMSProp, or Momentum (loss improvement). Positive values indicate performance better than baseline. We show training and validation losses for the outer-testing task distribution. On the majority of tasks, the learned optimizers outperform the baseline.

## 2 UNROLLED OPTIMIZATION FOR LEARNING OPTIMIZERS

### 2.1 PROBLEM FRAMEWORK

Our goal is to learn an optimizer which is well suited to some set of target optimization tasks. Throughout the paper, we will use the notation defined in Figure 2. Learning an optimizer can be thought of as a bi-level optimization problem (Franceschi et al., 2018), with *inner* and *outer* levels. The inner minimization consists of optimizing of the weights ($w$) of a target problem $\ell(w)$ by the repeated application of an update rule. The update rule is a parameterized function that defines how to map the weights at iteration $t$ to iteration $t + 1$: $w^{(t+1)} = u(w^{(t)}; \theta)$. Here, $\theta$ represents the parameters of the learned optimizer. In the outer loop, these optimizer parameters ($\theta$) are updated so as to minimize some measure of optimizer performance, the outer-objective $L(\theta)$. Our choice

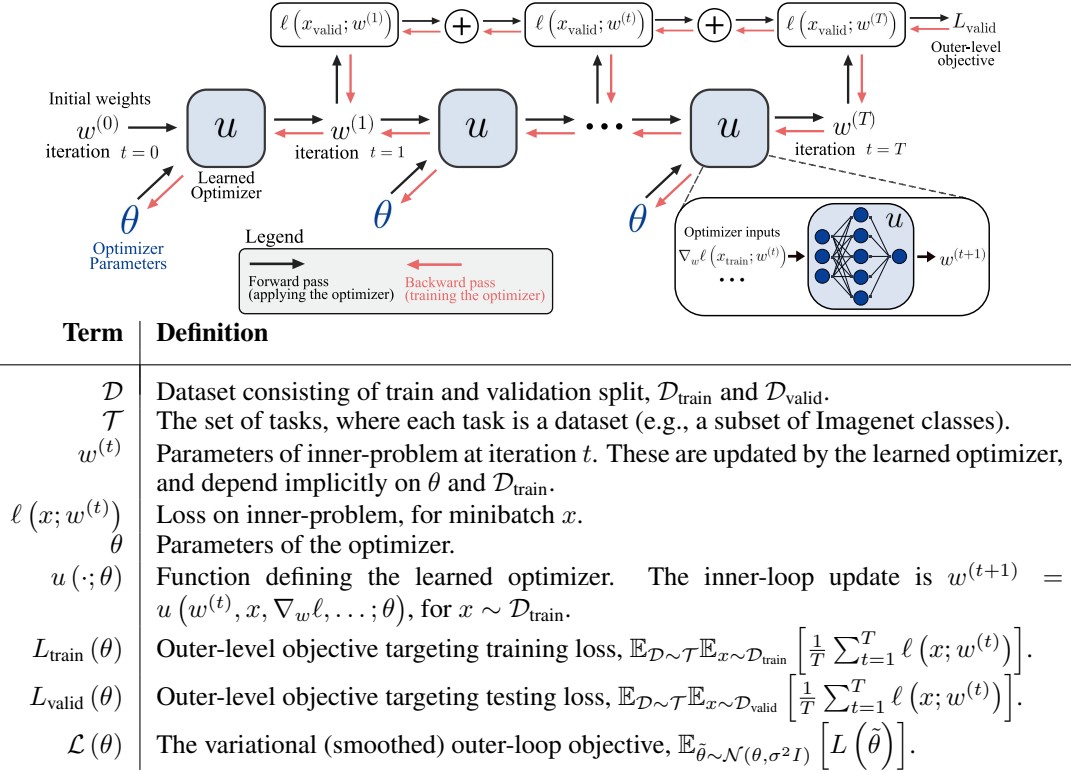

| Term | Definition |
|---|---|
| $\mathcal{D}$ | Dataset consisting of train and validation split, $\mathcal{D}_{\text{train}}$ and $\mathcal{D}_{\text{valid}}$. |
| $\mathcal{T}$ | The set of tasks, where each task is a dataset (e.g., a subset of Imagenet classes). |
| $w^{(t)}$ | Parameters of inner-problem at iteration $t$. These are updated by the learned optimizer, and depend implicitly on $\theta$ and $\mathcal{D}_{\text{train}}$. |
| $\ell\left(x; w^{(t)}\right)$ | Loss on inner-problem, for minibatch $x$. |
| $\theta$ | Parameters of the optimizer. |
| $u\left(\cdot; \theta\right)$ | Function defining the learned optimizer. The inner-loop update is $w^{(t+1)} = u\left(w^{(t)}, x, \nabla_w \ell, \ldots; \theta\right)$, for $x \sim \mathcal{D}_{\text{train}}$. |
| $L_{\text{train}}\left(\theta\right)$ | Outer-level objective targeting training loss, $\mathbb{E}_{\mathcal{D} \sim \mathcal{T}} \mathbb{E}_{x \sim \mathcal{D}_{\text{train}}} \left[ \frac{1}{T} \sum_{t=1}^{T} \ell\left(x; w^{(t)}\right) \right]$. |
| $L_{\text{valid}}\left(\theta\right)$ | Outer-level objective targeting testing loss, $\mathbb{E}_{\mathcal{D} \sim \mathcal{T}} \mathbb{E}_{x \sim \mathcal{D}_{\text{valid}}} \left[ \frac{1}{T} \sum_{t=1}^{T} \ell\left(x; w^{(t)}\right) \right]$. |
| $\mathcal{L}\left(\theta\right)$ | The variational (smoothed) outer-loop objective, $\mathbb{E}_{\tilde{\theta} \sim \mathcal{N}(\theta, \sigma^2 I)} \left[ L\left(\tilde{\theta}\right) \right]$. |

Figure 2: **Top:** Schematic of unrolled optimization. **Bottom** Definition of terms used in this paper.

for $L$ will be the average value of the target loss $\ell$ measured over either training or validation data. Throughout the paper, we use *inner-* and *outer-* prefixes to make it clear when we are referring to applying a learned optimizer on a target problem (inner) versus training a learned optimizer (outer).

## 2.2 UNROLLED OPTIMIZATION

In order to train an optimizer, we wish to compute derivatives of the outer-objective $L$ with respect to the optimizer parameters, $\theta$. Doing this requires unrolling the optimization process. That is, we can form an unrolled computational graph that consists of iteratively applying an optimizer ($u$) to optimize the weights ($w$) of a target problem (Figure 2). Computing gradients for the optimizer parameters involves backpropagating the outer loss through this unrolled computational graph. This is a costly operation, as the entire inner-optimization problem must be unrolled in order to get a single outer-gradient. Partitioning the unrolled computation into separate segments, known as truncated backpropagation, allows one to compute multiple outer-gradients over shorter segments. That is, rather than compute the full gradient from iteration $t = 0$ to $t = T$, we compute gradients in windows from $t = a$ to $t = a + \tau$. Below, we analyze two key problems with gradients computed via truncated unrolled optimization: bias and exploding norm.

## 2.3 EXPONENTIAL EXPLOSION OF GRADIENTS WITH INCREASED SEQUENCE LENGTH

We can illustrate the problem of exploding gradients analytically with a simple example: learning a learning rate. Following the notation in Figure 2, we define the optimizer as

$$w^{(t+1)} = u(w^{(t)}; \theta) = w^{(t)} - \theta \nabla \ell\left(w^{(t)}\right),$$

where $\theta$ is a scalar learning rate that we wish to learn for minimizing some target problem $\ell(w^{(t)})$. For simplicity, we assume no minibatch $x$.

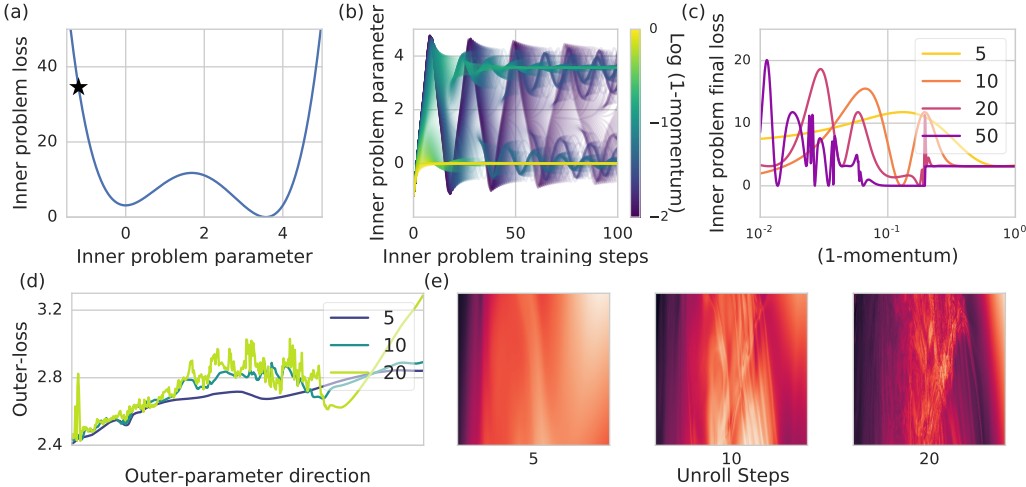

Figure 3: Outer-problem optimization landscapes become increasingly pathological with increasing inner-problem step count. **(a)** A toy 1D inner-problem loss surface with two local minimum. Initial parameter value $w^{(0)}$ is indicated by the star. **(b)** Final inner-parameter value $w^{(T)}$ as a function of the number of inner problem training steps $T$, when inner-problem training is performed by SGD+momentum. Color denotes different values of the optimizer's momentum parameter. Low momentum values converge to the first local minimum at $w = 0$. Slightly higher momentum values escape this minimum to settle at the global minimum ($w \approx .5$). Even larger values oscillate before eventually settling in one of the two minimum. **(c)** The final loss after some number steps of optimization as a function of the momentum. The final loss surface is smooth for small number of training steps $T$. However, larger values of $T$ result in near discontinuous loss surfaces around the transition points between the 2 minimum. **(d)** Similar to **c**, where the inner problem is a MLP, the learned optimizer is the one used this paper, and for a 1D slice through the outer-parameters $\theta$ along the gradient direction. **(e)** 2D rather than 1D slices through $\theta$, for different numbers of inner-loop steps. Intensity indicates value of $L_{\text{train}}(\theta)$. Similar pathologies are observed to those which manifest in the toy problem.

The quantity we are interested in is the outer-gradient, the derivative of the loss after $T$ steps of gradient descent with respect to $\theta$. We can compute this outer-gradient (see Appendix A) as:

$$\frac{d\ell(w^{(T)})}{d\theta} = \left\langle -g^{(T)}, \sum_{i=0}^{T-1} \left( \prod_{j=1}^{T-1} (I - \theta H^{(j)}) \right) g^{(i)} \right\rangle,$$

where $g^{(i)}$ and $H^{(j)}$ are the gradient and Hessian of the target problem $\ell(w)$ at iteration $i$ and $j$, respectively. We see that this equation involves a sum of products of Hessians. In particular, the first term in the sum involves a product over the *entire sequence* of Hessians observed during training. That is, the outer-gradient becomes a matrix polynomial of degree $T$, where $T$ is the number of gradient descent steps. Thus, the outer-gradient grows exponentially with $T$.

We can see another problem with long unrolled gradients empirically. Consider the task of optimizing a loss surface with two local minima defined as $\ell(w) = (w-4)(w-3)w^2$ with initial condition $w^{(0)} = -1.2$ using a momentum based optimizer with a parameterized momentum value $\theta$ (Figure 3a). At low momentum values the optimizer converges in the first of the two local minima, whereas for larger momentum values the optimizer settles in the second minima. With even larger values of momentum, the iterate oscillates between the two minima before settling. We visualize both the trajectory of $w^{(t)}$ over training and the final loss value for different momentum values in Figure 3b and 3c. With increasing unrolling steps, the loss surface as a function of the momentum $\theta$ becomes less and less smooth, and develops near-discontinuities at some values of the momentum.

In the case of neural network inner-problems and neural network optimizers, the outer-loss surface can grow even more complex with increasing number of unrolling steps. We illustrate this in Figure

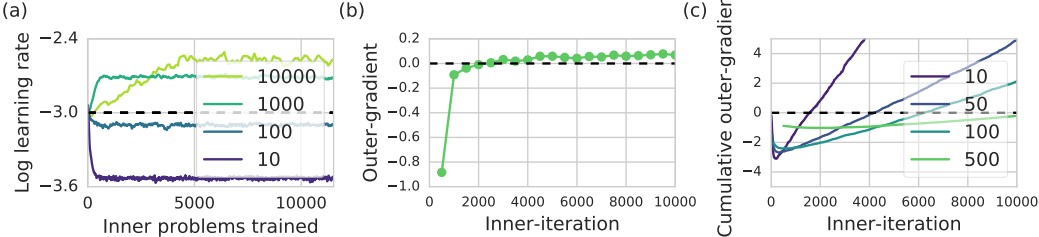

Figure 4: Large biases can result from reducing the number of steps per truncation in unrolled optimization. **(a)** Each line represents an experiment with the same total number of inner-steps (10k) but a different number of unrolling steps per truncation. In all cases, the initial learning rate is $1e-3$ (dashed line). We find low truncation amounts move away from the optimal learning rate ($\sim 4e-3$). **(b)** Outer-gradient at each truncation over the course of inner-training. Initial outer-gradients are highly negative, trying to increase the learning rate, while later outer-gradients are slightly above zero, decreasing learning rate. The total outer-gradient is the sum of these two competing directions. **(c)** Cumulative value of outer-gradient vs inner-training step. When a line crosses the zero point, the outer-learned learning rate is at an equilibrium. For low unrolling steps, the cumulative sum is positive after a small number of steps, resulting in a increasing learning rate, and decreasing outer-loss. As more inner-steps are taken, the bias increases, eventually flipping the outer-gradient direction resulting in increasing outer-loss.

3d and 3e for slices through the loss landscape $L(\theta)$ of the outer-problem for a neural network optimizer.

## 2.4 INCREASING BIAS WITH TRUNCATED GRADIENTS

Here, we demonstrate the bias introduced by using a short truncation window in unrolled optimization. These results are similar to those presented in Wu et al. (2016), except that we utilize *multiple* truncations rather than a single, shortened unroll. First, consider outer-learning the learning rate of Adam when optimizing a small two layer neural network on MNIST(LeCun, 1998). Grid search can be used to find the optimal learning rate which is $0.004$. We initialize Adam a learning rate of $0.001$ and outer-train using increasing truncation amounts (Figure 4a). Despite initializing close to the optimal learning rate, when outer-training with truncated backprop the resulting learning rate decreases – that is the sum of truncated outer-gradients are anti-correlated to the true outer-gradient. We visualize the per-truncation gradients of 500 steps in Figure 4b and cumulative truncated gradients in Figure 4c. Early in inner-training there is a large negative outer-gradient which increases the learning rate. Later in inner-training, the outer-gradients are positive, decreasing the learning rate. Thus, the optimizer parameter is pulled in *opposite* directions by truncated gradients early versus late in inner-training, revealing an inherent tension with truncated unrolled optimization.

## 3 TOWARDS STABLE TRAINING OF LEARNED OPTIMIZERS

To perform outer-optimization of a loss landscape with high frequency structure like that in Figure 3, one might intuitively want to smooth the outer-objective loss surface. To do this, instead of optimizing $L(\theta)$ directly, we instead optimize a smoothed outer-loss $\mathcal{L}(\theta)$,

$$\mathcal{L}(\theta) = \mathbb{E}_{\tilde{\theta} \sim \mathcal{N}(\theta, \sigma^2 I)} \left[ L\left(\tilde{\theta}\right) \right],$$

where $\sigma^2$ is a fixed variance (set to 0.01) which determines the degree of smoothing. This is the same approach taken in variational optimization (Staines & Barber, 2012). We can construct two different unbiased gradient estimators for $\mathcal{L}(\theta)$: one via the reparameterization trick (Kingma & Welling, 2013), and one via the log-derivative trick similar to what is done in evolutionary strategies (ES) (Wierstra et al., 2008). We denote the two estimates as $g_{\text{rp}}$ and $g_{\text{es}}$ respectively,

$$g_{\text{rp}} = \mathbb{E}_{e \sim N(0, I)} \left[ \nabla_\theta L\left(\theta + e\sigma\right) \right],$$

$$g_{\text{es}} = \mathbb{E}_{\hat{\theta} \sim N(\theta, \sigma^2 I)} \left[ L\left(\tilde{\theta}\right) \nabla_\theta [log\left( N\left(\tilde{\theta}; \theta, \sigma^2\right)\right)] \right].$$

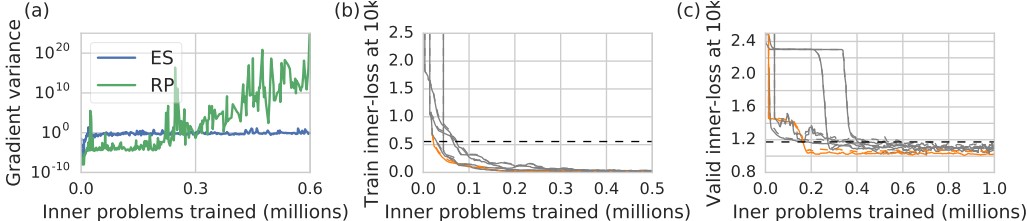

Figure 5: **(a)** As outer-training progresses, the variance of the reparameterization gradient estimator grows, while the variance of the ES estimator remains constant. **(b,c)** Performance after 10k iterations of inner-training for different outer-parameters over the course of outer-training. Each line represents a different random initialization of outer-parameters ($\theta$). Models are trained targeting the train outer-objective (b), and the validation outer-objective (c). Dashed lines indicate performance of hyperparameter tuned Adam. Model's in orange are the best performing and used in §4.

Following the insight from (Parmas et al., 2018) in the context of reinforcement learning[1], we combine these estimates using inverse variance weighting (Fleiss, 1993),

$$g_{\text{merged}} = \frac{g_{\text{rp}}/\sigma_{\text{rp}}^{-2} + g_{\text{es}}/\sigma_{\text{es}}^{-2}}{\sigma_{\text{rp}}^{-2} + \sigma_{\text{es}}^{-2}}, \tag{1}$$

where $\sigma_{\text{rp}}^2$ and $\sigma_{\text{es}}^2$ are the empirical variances of $g_{rp}$ and $g_{es}$ respectively. When outer-training learned optimizes we find the variances of $g_{\text{es}}$ and $g_{\text{rp}}$ can differ by as many as 20 orders of magnitude (Figure 5a). This merged estimator addresses this by having at most the lowest of the two variances. To further reduce variance, we use antithetic sampling when estimating $g_{es}$.

The cost of computing a single sample of $g_{\text{es}}$ and $g_{\text{rp}}$ is thus 2 forward and backward passes of an unrolled optimization. To compute the empirical variance, we leverage data parallelism to compute multiple samples of $g_{\text{es}}$ and $g_{\text{rp}}$. In theory, the samples used to evaluate $\sigma_{\text{rp}}^2$, $\sigma_{\text{es}}^2$ must be independent of $g_{\text{es}}$ and $g_{\text{rp}}$, but in practice we found good performance using the same samples for both. Finally, an increasing curriculum over steps per truncation is used over the course of outer-training. This introduces bias early in training, but also allows for far more frequent outer-weight updates, resulting in much faster outer-training in terms of wall-clock time. The full outer-training algorithm is described in Appendix B.

## 4 EXPERIMENTS

### 4.1 OPTIMIZER ARCHITECTURE

The optimizer architecture used in all experiments consists of a simple fully connected network, with one hidden layer containing 32 ReLU units. This network is applied to each target problem weight independently. The outputs of the MLP consist of an update direction and a log learning rate, which are combined to produce weight updates. The network for each weight takes as input: the gradient with respect to that weight, parameter value, RMS gradient terms (Lucas et al., 2018), exponentially weighted moving averages of gradients at multiple time scales, as well as a representation of the current iteration. Many of these input features were motivated by Wichrowska et al. (2017). We conduct ablation studies for these inputs in §4.5. See Appendix C for further architectural details.

### 4.2 OPTIMIZER TARGET PROBLEM

The problem that each learned optimizer is trained against consists of training a three layer convolutional neural network (32 units per layer) inner-trained for ten thousand inner-iterations on 32x32x3 image classification tasks. We split the Imagenet dataset (Russakovsky et al., 2015) by class into

---

[1] Parmas et al. (2018) go on to propose a more sophisticated gradient estimator that operates on a per iteration level. While this should result in an even lower variance estimator in our setting, we find that the simpler solution of combing both terms at the end is easier to implement and works well in practice.

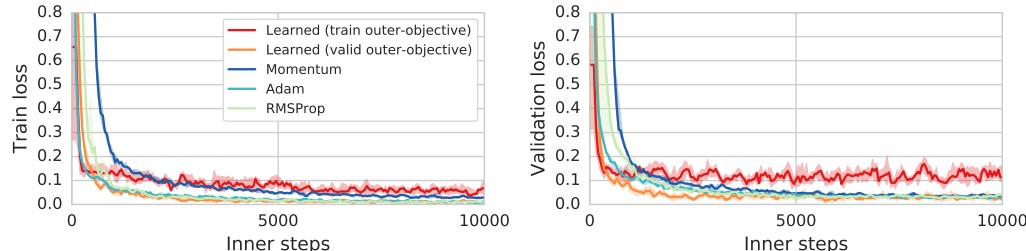

Figure 6: Despite outer-training only targeting optimizer performance on a three layer convolutional network trained on 10 way classification problems derived from 32x32 Imagenet, the learned optimizers demonstrate some ability to generalize to new architectures and datasets. Plot shows train and validation performance on a six layer convolutional neural network trained on 28x28 MNIST. We compare against hyperparameter tuned Adam, RMSProp, and SGD+Momentum. Note that the optimizer trained to target validation loss generalizes better than the one trained to target train loss. See Appendix E for experiments testing generalization to additional tasks.

700 training and 300 validation classes, and sample training and validation problems by sampling 10 classes at random using all images from each class. This experimental design lets the optimizer learn problem specific structure (e.g. convolutional networks trained on object classification), but does not allow the optimizer to memorize particular weights for the base problem. See Appendix C for further details.

### 4.3    OUTER-TRAINING

To train the optimizer, we linearly increase the number of unrolled steps from 50 to 10,000 over the course of 5,000 outer-training weight updates. The number of unrolled steps is additionally jittered by a small percentage (sampled uniformly up to 20%). Due to the heterogeneous, small workloads, we train with asynchronous batched SGD using 128 CPU workers.

Figure 5 shows the performance of the optimizer (averaged over 40 randomly sampled outer-train and outer-test inner-problems) while outer-training. Despite the stability improvements described in the last section, there is still variability in optimizer performance over random initializations of the optimizer parameters. We use training loss to select the best model and use this in the remainder of the evaluation.

### 4.4    LEARNED OPTIMIZER PERFORMANCE

Figure 1 shows performance of the learned optimizer, after outer-training, compared against other first-order methods on a sampled validation task (classes not seen during outer-training). For all first-order methods, we report the best performance after tuning the learning rate by grid search using 11 values over a logarithmically spaced range from $10^{-4}$ to 10. When outer-trained against the training outer-objective, $L_{\text{train}}$, we achieve faster convergence on training loss by a factor of 5x (Figure 1a), but poor performance on validation loss (Figure 1b). However, when outer-trained against the validation outer-objective, $L_{\text{valid}}$, we also achieve faster optimization *and* reach a lower validation loss (Figure 1b).

Figure 1 summarizes the performance of the learned optimizer across many sampled validation tasks. It shows the difference in final validation loss between the *best* first-order method and the learned optimizer. We choose the best first-order method by selecting the best validation performance over RMSProp, SGD+Momentum, and Adam. This learned optimizer (which does not require tuning on the validation tasks) outperforms the best baseline optimizer 98% of the time.

Although the focus of our approach was not generalization, we find that our learned optimizer nonetheless generalizes to varying degrees to dissimilar datasets, different numbers of units per layer, different number of layers, and even to fully connected networks. In Figure 6 we show performance on a six layer convolutional neural network trained on MNIST. Despite the different number of layers, different dataset, and different input size, the learned optimizers still reduces the loss,

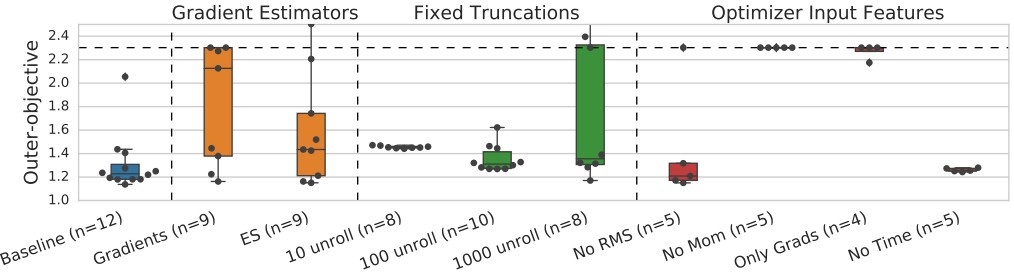

Figure 7: We compare the model described in §4.1 with different features removed. Shown above is the distribution of outer-loss performance across random seed averaged between 380k and 420k inner problem trained. For full learning curves, see Appendix G. First, with regard to gradient estimators we find the combined estimator is more stable and out performs both the analytic gradient, and evolutionary strategies. With regard to a curriculum, we find poor convergence with lower numbers of unrolling steps (10, 100) and high variance with larger numbers of truncation's steps (1000). Finally with regard to optimizer features, we find our learned optimizers perform nearly as well *without* the RMS terms and *without* the time horizon input, but fails to converge when not given access to momentum based features.

and in the case of the validation outer-objective trains faster to a lower validation loss. We further explore the limits of generalization of our learned optimizer on additional tasks in Appendix E.

## 4.5 ABLATIONS

To assess the importance of the gradient estimator discussed in §3, the unrolling curriculum §4.3, as well as the features fed to the optimizer enumerated in §4.1, we re-trained the learned optimizer removing each of these additions. In particular, we trained optimizers with: only the analytic gradient (Only Gradients), only with evolutionary strategies (Only ES), a fixed number unrolled steps (10, 100, 1000) as opposed to a schedule, no RMS gradient scaling (No RMS), no momentum terms (No Mom), no momentum and no RMS scaling (Only Grads), and without the current iteration (No Time). Figure 7 summarizes these findings, showing the learned optimizer performance for each of these ablations. We find that the gradient estimator (in §3) and a increasing schedule of unroll steps are critical to performance, along with including momentum as an input to the optimizer.

## 5 DISCUSSION

In this work we demonstrate two difficulties when training learned optimizers: "exploding" gradients, and a bias introduced by truncated backpropagation through time. To combat this, we construct a variational bound of the outer-objective and minimize this via a combination of reparameterization and ES style gradient estimators. By using our combined estimator and a curriculum over truncation step we are able to train learned optimizers that achieve more than five times speedup on wallclock time as compared to existing optimizers.

In this work, we focused on applying optimizers to a restricted family of tasks. While useful on its own right (e.g. rapid retraining of models on new data), future work will explore the limits of "no free lunch" (Wolpert & Macready, 1997) to understand how and when learned optimizers generalize across tasks. We are also interested in using these methods to better understand what problem structure our learned optimizers exploit. By analyzing the trained optimizer, we hope to develop insights that may transfer back to hand-designed optimizers. Outside of meta-learning, we believe the gradient estimator presented here can be used to train other long time dependence recurrent problems such as neural turning machines (Graves et al., 2014), or neural GPUs (Kaiser & Sutskever, 2015).

Much in the same way deep learning has replaced feature design for perceptual tasks, we see meta-learning as a tool capable of learning new and interesting algorithms, especially for domains with unexploited problem-specific structure. With better outer-training stability, we hope to improve our ability to learn interesting algorithms, both for optimizers and beyond.

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

## A  DERIVATION OF THE UNROLLED GRADIENT

After $T$ steps of gradient descent, the final loss becomes $\ell(w^{(T)})$. We can update the parameters of the optimizer ($\theta$) by computing the derivative of this final loss with respect to $\theta$: $\frac{d\ell(w^{(T)})}{d\theta} = \nabla\ell(w^{(T)})^T \frac{dw^{(T)}}{d\theta}$. The second term in this expression can be defined recursively as:

$$\frac{dw^{(T)}}{d\theta} = \left(I - \theta\nabla^2\ell\left(w^{(T-1)}\right)\right)\frac{dw^{(T-1)}}{d\theta} - \nabla\ell\left(w^{(T-1)}\right),$$

where $\nabla\ell$ and $\nabla^2\ell$ are the gradient and Hessian of the target problem, respectively. This expression highlights where the exploding outer-gradient comes from: the recursive definition of $\frac{dw^{(T)}}{d\theta}$ means that computing it will involve a product of the Hessian at every iteration.

## B  OUTER-TRAINING ALGORITHM

Initialize outer-parameters ($\theta$).
**while** Outer-training, for each parallel worker **do**
    Sample a dataset $\mathcal{D}$, from $\mathcal{T}$.
    Initialize the inner loop parameters $w^{(0)}$ randomly.
    **for** Each truncation in the inner loop **do**
        Sample ES perturbation: $e \sim N(0, \sigma^2 I)$.
        Sample a number of steps per truncation, $k$, based on current outer-training index.
        Compute a positive, and negative sequence starting from $w^{(t)}$ by iteratively applying (for $k$ steps), $u(\cdot; \theta + e)$, to $w^{(t)}$
        Compute a pair of outer-objectives $L^+$, and $L^-$ using the 2 sequences of $w$ from $t$ to $k$ using either the train or validation inner-problem data.
        Compute a single sample of $g^{rp} = \nabla_\theta \frac{1}{2}(L^+ + L^-)$.
        Compute a single sample of $g^{es} = \frac{1}{2}(L^+ - L^-)\nabla_\theta log(N(e; \theta, \sigma^2 I))$
        Store the sample of $(g_{rp}, g_{es})$ in a buffer until a batch of samples is ready.
        Assign the final $w$ from one of the two sequences to $w^{(t+k)}$.
    **end for**
**end while**
**while** Outer-training and in parallel **do**
    When a batch of gradients is available, compute empirical variance and empirical mean of each weight for each estimator.
    Use equation 1 to compute the combined gradient estimate.
    update meta-parameters with SGD: $\theta \leftarrow \theta - \alpha g_{combined}$ where $\alpha$ is a learning rate.
**end while**
      **Algorithm 1:** Outer-training algorithm using the combined gradient estimator.

## C  ARCHITECTURE DETAILS

### C.1  ARCHITECTURE

In a similar vein to diagonal preconditioning optimizers, and existing learned optimizers our architecture operates on each parameter independently. Unlike other works, we do not use a recurrent model as we have not found applications where the performance gains are worth the increased computation. We instead employ a single hidden layer feed forward MLP with 32 hidden units. This MLP takes as input momentum terms, as well as rms terms inspired by Adam computed at a few different decay values: [0.5, 0.9, 0.99, 0.999, 0.9999] (Wichrowska et al., 2017). A similar idea has been explored with regard to momentum parameters in Lucas et al. (2018). We also pass in 2 terms: $r = \frac{1}{rms + \epsilon}$ and $rms$ also from computations performed in Adam. Despite being critical in (Wichrowska et al., 2017) we find these features of minimal impact (see §4.5). This is $5 \cdot 4 = 20$ features in total. The current gradient as well as the current weight value are also used as features.

By passing in weight values, the optimizer can learn to do arbitrary norm weight decay. To emulate learning rate schedules, the current training iteration is fed in transformed via applying a tanh squashing functions at different timescales: $tanh(t/\eta - 1)$ where $\eta$ is the timescale. We use 9 timescales logarithmicly spaced from (3, 300k). This leaves us in total with 31 features.

All non-time features are normalized by the second moment with regard to other elements in the "batch" dimension (the other weights of the weight tensor). We choose this over other normalization strategies to preserve directionality. These activations are then passed the into a hidden layer, 32 unit MLP with relu activations. Many existing optimizer hyperparameters (such as learning rate) operate on an exponential scale. As such, the network produces two outputs, and we combine them in an exponential manner: $exp(\lambda_{exp}o_1)\lambda_{lin}o_2$ making use of two temperature parameters $\lambda_{exp}$ and $\lambda_{lin}$ which are both set to $1e-3$. Without these scaling terms, the default initialization yields steps on the order of size 1 – far above the step size of any known optimizer and result in highly chaotic regions of $\theta$. It is still possible to optimize given our estimator, but training is slow and the solutions found are quite different.

## C.2 OUTER-DATA DISTRIBUTIONS

The outer-training set consists of a family of 10 way classification problems using cross entropy loss on subsets of 32x32 Imagenet. To form a outer-train and outer-test set, we randomly split imagenet into 700 classes for outer-train, and 300 classes for outer-test.

## C.3 INNER-PROBLEM

The optimizer targets a 3 layer convolutional neural network with 3x3 kernels, and 32 units per layer. The first 2 layers are stride 2, and the 3rd layer has stride 1. We use relu activations and glorot initializations (Glorot & Bengio, 2010). At the last convolutional layer, an average pool is performed, and a linear projection is applied to get the 10 output classes.

## C.4 OUTER-TRAINING

We train using the algorithm described in 1 using a linear schedule on the number of unrolling steps from 50 - 10k over the course of 5k outer-training iterations. To add variation in length, we additionally shift this length by a percentage uniformly sampled between (-20%, 20%). We optimize the outer-parameters, $\theta$, using Adam (Kingma & Ba, 2014) with a batch size of 128 and with a learning rate of 0.003 for the training outer-objective and 0.0003 for the validation outer-objective, and $\beta_1 = 0.5$(following existing literature on non-stationary optimization (Arjovsky et al., 2017)). While both values of learning rate work for both outer-objectives, we find the validation outer-objective to be *considerably* harder, and training is more stable with the lower learning rate.

# D  ADDITIONAL INNER LOOP PROBLEM LEARNING CURVES

We plot additional learning curves from both the outer-train task distribution and the outer-validation task distribution. The horizontal lines represent the minimum performance achieved over 20k steps. See Figure 1.

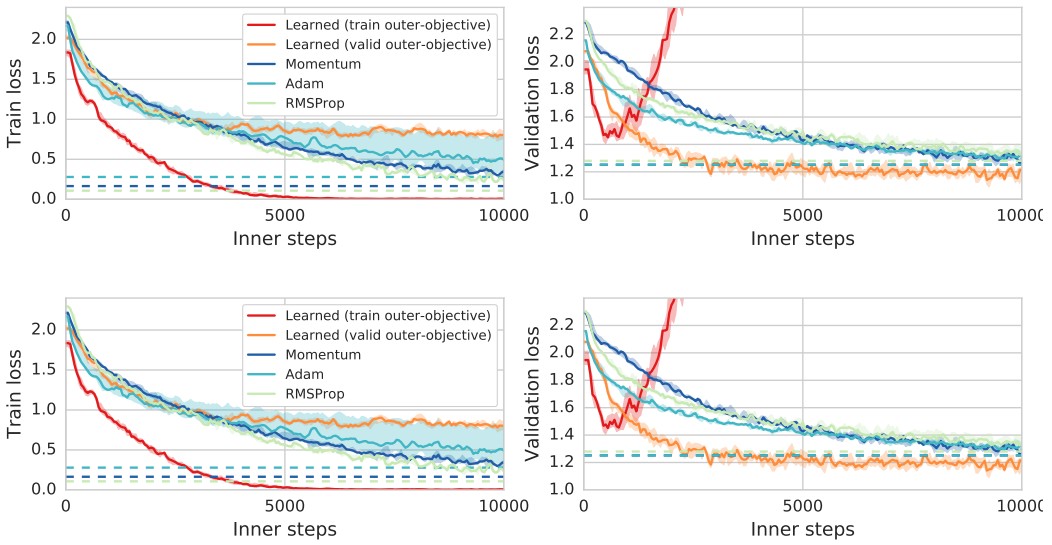

Figure 8: Additional outer-validation problems.

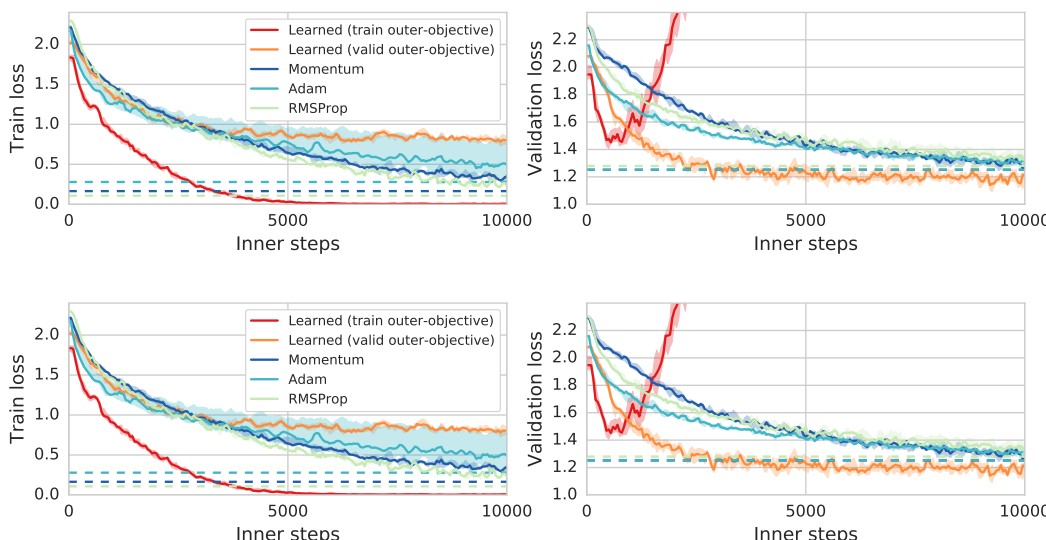

Figure 9: Outer-training problems.

# E OUT OF DOMAIN GENERALIZATION

In this work, we focus our attention to learning optimizers over a specific task distribution (3 layer convolutional networks trained on 10 way subsets of 32x32 Imagenet). In addition to testing on these in domain problems (Appendix D), we test our learned optimizer on a variety of out of domain target problems. Despite little variation in the outer-training task distribution, our models show promising generalization when transferred to a wide range of different architectures (fully connected, convolutional networks) depths (2 layer to 6 layer) and number of parameters (models roughly 16x more parameters). We see these as promising sign that our learned optimizer has a reasonable but not perfect inductive bias. We leave training with increased variation to encourage better generalization as an area for future work.

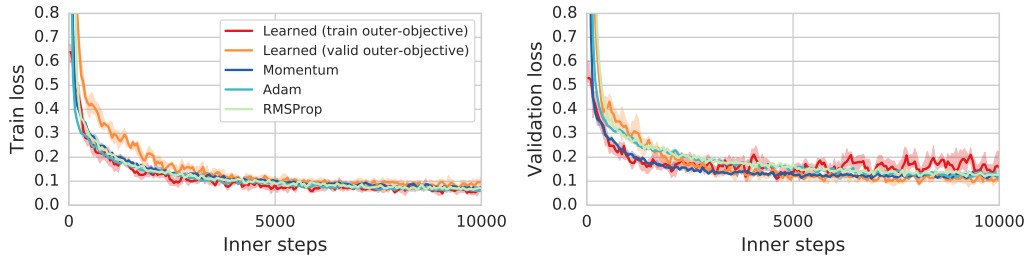

Figure 10: Inner problem: 2 hidden layer fully connected network. 32 units per layer with relu activations trained on 14x14 MNIST.

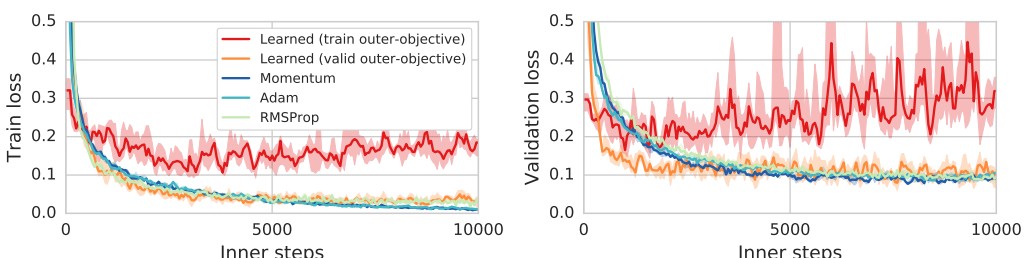

Figure 11: Inner problem: 3 hidden layer fully connected network. 128 units per layer with relu activations trained on 14x14 MNIST.

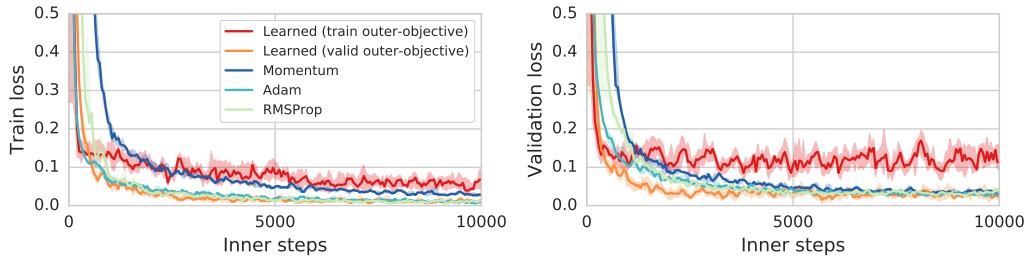

Figure 12: Inner problem: 6 convolutional layer network. 32 units per layer, strides: [2,1,2,1,1,1] with relu activations on 28x28 MNIST.

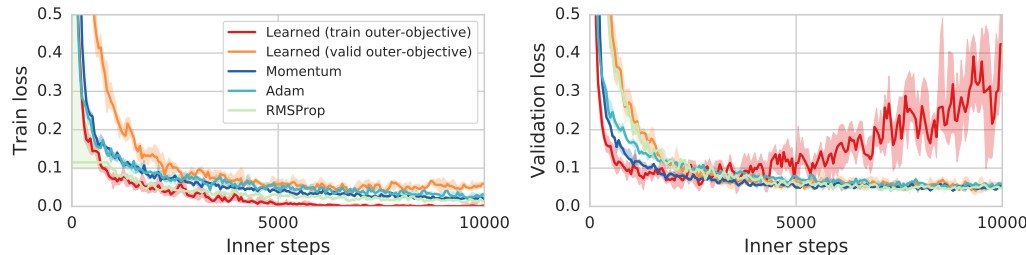

Figure 13: Inner problem: 3 convolutional layer network. 32 units per layer, strides: [2,2,1] with relu activations on 28x28 MNIST.

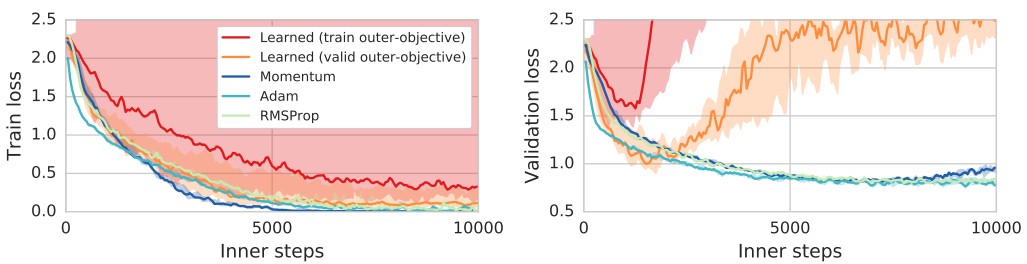

Figure 14: Inner problem: 3 convolutional layer network. 128 units per layer, strides: [2,2,1] with relu activations trained on a 10 way classification sampled from 32x32 Imagenet (using holdout classes).

## F  INNER-LOOP TRAINING SPEED

When training models, often one cares about taking less wallclock time as compared to performance per weight update. Much like existing first order optimizers, the computation performed in our learned optimizer is linear in terms of number of parameters in the model being trained and smaller than the cost of computing gradients. The bulk of the computation in our model consists of two batched matrix multiplies of size 31x32, and 32x2. When training models that make use of weight sharing, e.g. RNN or CNN, the computation performed per weight often grows super linearly with model size. As the learned optimizer methods are scaled up, the additional overhead in performing more complex weight updates will vanish.

For the specific models we test in this paper, we measure the performance of our optimizer on CPU and GPU. We implement Adam, SGD, and our learned optimizer in TensorFlow for this comparison. Given the small scale of problem we are working at, we implement training in graph in a *tf.while_loop* to avoid TensorFlow Session overhead. We use random input data instead of real data to avoid any data loading confounding. On CPU the learned optimizer executes at 80 batches a second where Adam runs at 92 batches a second and SGD at 93 batches per second. The learned optimizer is 16% slower than both.

On a GPU (Nvidia Titan X) we measure 177 batches per second for the learned and 278 batches per second for Adam, and 358 for sgd. This is or 57% slower than adam and 102% slower than SGD.

Overhead is considerably higher on GPU due to the increased number of ops, and thus kernel executions, sent to the GPU. We expect a fused kernel can dramatically reduce this overhead. Despite the slowdown in computation, the performance gains (greater than 400% faster in steps) far exceed the slowdown, resulting in an optimizer that is still considerably faster when measured in wallclock time.

# G  ABLATION LEARNING CURVES

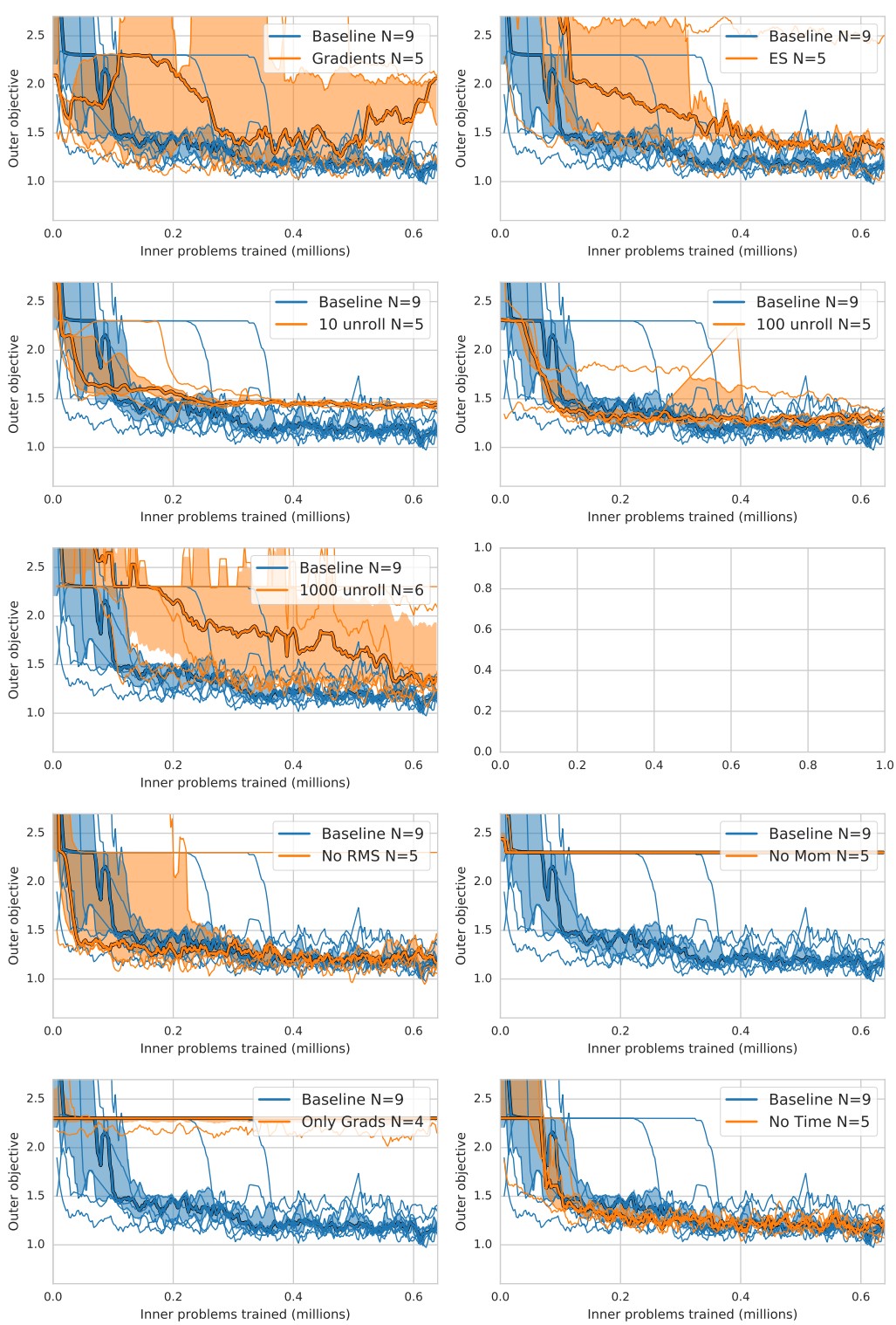

Figure 15: Training curves for ablations described in §4.5. The thick line bordered in black is the median performance, with the shaded region containing the 25% and 75% percentile. Thinner solid lines are individual runs.

