# OpenReview forum: "Learned optimizers that outperform on wall-clock and validation loss"
_ICLR.cc/2019/Conference_

### Official Review · AnonReviewer1 · 2018-10-15
**Good idea, but needs more work**

**Rating:** 5
**Confidence:** 4

**Review:**

Summary:
The paper presents a method for "learning an optimizer"(also in the literature Learning to Learn and a form of Meta-Learning) by using a Variational Optimization for the "outer" optimizer loss. The mean idea of the paper is to combine both the reparametrized gradient and the score-function estimator for the Variational Objective and weight them using a product of Gaussians formula for the mean. The method is simple and clearly presented. The paper also presents issues with the standard "learning to learn" optimizers, one being the short-horizon bias and as credited by the authors has been observed before in the literature, and the second one is what is termed the "exponential explosion of gradients" which I think lacks enough justification as currently presented (see below for details). The ideas are clearly stated, although the work is not groundbreaking, but more on combining several ideas into a single one.

Experiments:
The authors evaluate their method on a single task which consists of optimizing a 3-layer convolutional neural network on downsampled images from ImageNet. A key idea, not new to this work, is to optimize the meta-optimizer with respect to the validation dataset rather than the training, which seems to be crucial for any meaningful training to happen. Although the experiments do show so promising results, they seem to be somewhat limited (see below for details). There is also a small ablation study on how do different features presented to the optimizer affect its performance. Given the still small-scale experiments, I'm not sure this is a significant result for the community.

Conclusion:
As a whole, I think the idea in the paper is a good one and worth investigating further. However, the objections I have on section 2.3 and the experiments seem to indicate that there needs to be more work into this paper to make it ready for publication.


On section 2.3 and the explosion of gradients:

There is a mistake in the equation on page 4 regarding the "gradient with respect to the learning rate". Although the derivation in Appendix A is correct, the inner product in the equation starts wrongly from j=0, where it should in fact start at j = i + 1. To be more clear the actual enrolled equation for dw^T/dt for 3 steps back is:

dw^T/dt = (I - tH^{T-1})(I - tH^{T-2})(I - tH^{T-3}) dw^{T-3} - (I - tH^{T-1})(I - tH^{T-2}) g^{T-3} - (I - tH^{T-1}) g^{T-2} - g^{T-1}

Hence the product must start at j = i + 1.
It is correct that in this setting the equation is a polynomial of degree T of the Hessian, however, there are several important factors that the authors have not discussed. Namely, if the learning rate is chosen accordingly such that the spectral radius of the Hessian is less than 1/t then rather than the gradient exploding the higher order term will vanish. However, even if they do vanish for large T since the Hessian plays with smaller and smaller power to more recent gradients (after correcting the mistake in the equation) than the actual T-step gradient will never vanish (in fact even if tH = I then dw^T/dt = g^{T-1}). Hence the claims of exploding gradients made in this section coupled with the very limited theoretical analysis seem to unconvincing that this is nessacarily an issue and under what circumstances they are.

The toy example with l(w) = (w - 4)(w - 3) w^2 is indeed interesting for visualizing a case where the gradient explosion does happen. However, surprisingly here the authors rather than optimizing the learning rate, which they analyzed in the previous part of the section, they are now optimizing the momentum. The observation that at high momentum the training is unstable are not really surprising as there are fundamental reasons why too high momentum leads to instabilities and these have been analyzed in the literature. Additionally, it is not mentioned what learning rate is used, which can also play a major role in the effects observed.

As a whole, although the example in this section is interesting, the claims made by the authors and some of the conclusions seem to lack any significant justifications, in addition to the fact that usually large over-parameterized models behave differently than small models.


Experiments:

I have a few key issues with the experimental setup, which I think need to be addressed:

1. The CNN being optimized is quite small - only 3 layers. This allows the authors to train everything on a CPU. The key issue here, as well with previous work on Learning to Learn, is that it is not clear how scalable is this method to very Deep Networks.

2. Figure 1 - The setup is to optimize the problem for 10000 iterations, however, I think it is pretty clear even to the naked eye that the standard first-order optimizers (Adam/RMS/Mom) have not fully converged on this problem. Hence I think its slightly unfair to compare their "final performance" after this fixed period. Additionally using the curriculum the "meta"-optimizer is trained explicitly for 10000 iterations. Hence, it is also unclear if it retains its stability after letting it run for longer. From the text it is also unclear whether the authors have optimized the parameters of the first-order methods with respect to their training or validation performance - I hope this is the latter as that is the only way to fairly compare the two approaches.

3. Figure 6 - the results here seem to indicate that the learned optimizer transfers reasonably well, achieving similar performance to first-order methods (slightly faster validation reduction). Given however that these are plots for only 10000 iterations it is still unclear if this is scalable to larger problems.

---

> ### Author Response · Authors · 2018-11-14
> **Thank you**
>
> Thank you for your thoughtful review! We will take these under consideration for a future submission. Comments addressed below.
>
> j=i+1: You are indeed correct. This was an unfortunate typo we realized just after submitting. Good catch and thank you for giving our work such a thorough review!
>
> Your points as to exploding or vanishing gradients are correct. If the optimal learning rate is known ahead of time (requiring knowledge of the hessian at each iteration) it is possible to not have exploding gradients. In practice, however, it is rarely possible to find useful bounds on the eigenvalues of the Hessian in neural network training. This, coupled with the fact that the optimal learning rate is often at the edge of unstable dynamics, can lead to learning rates in the unstable regime and thus exploding gradients.
>
> We appreciate the comment about vanishing gradients. We will update this section discussing that gradients do not vanish -- only explode. (Note that the outer-parameters are used at every training iteration -- so even if the backpropagated outer-gradient shrinks exponentially with respect to unrolled optimization steps, it does not vanish with respect to the outer-parameters.)
>
> " However, surprisingly here the authors rather than optimizing the learning rate, which they analyzed in the previous part of the section, they are now optimizing the momentum."
>
> Here, we use momentum as it more clearly maps to a physical phenomenon, in the hope that it provides better intuition. Similar behavior holds for learning rate -- it is possible to take a step that is either just under or just over a local maximum, resulting in diverging trajectories.
>
> "in addition to the fact that usually large over-parameterized models behave differently than small models."
> We use these toy problems as a tool to build intuition as understanding the non-convex setting is very complex. That being said, we have done additional work (not included) around exploring these effects on larger problems. In particular, we are able to find multiple saddle points / paths to descend a loss function resulting in discontinuous trajectories, and exploding gradients (as in the toy case). We do this by taking 2d slices through the inner problems, and sweeping meta-parameters. We observe discontinuous jumps in final location and trajectory. We will look into adding, likely in the appendix, some examples of this behavior in large networks. We emphasize that Figure 3e already shows a slice through the outer-loss landscape for a neural network task and a neural network optimizer, and that pathological behavior in the unrolled loss landscape is visible in this figure.
>
> Scale: As of now, these methods are quite expensive. As a result, the field mostly explores meta-training on small scale tasks. Despite the expense, this work operates on considerably larger models than almost all prior work. We train conv-nets (as opposed to small MLPs) and train for 10k inner-iterations -- around 2 orders of magnitude longer than most existing work. We are extremely interested in pushing these methods further -- applying to even larger problems, but instability in meta-training has previously been a major obstacle to such scaling.
>
> "Hence I think its slightly unfair to compare their "final performance" after this fixed period.": When evaluating on training loss, we show optimization speed. With regard to validation loss, you are correct that we do not provide sufficient detail for the "better final performance" claim. In practice, we find overfitting occurs in under 20k inner-iterations (Shown in the dashed line on the figures). We will modify the text / figures to show the training step where existing optimizers begin to overfit.
>
> " From the text it is also unclear whether the authors have optimized the parameters of the first-order methods with respect to their training or validation performance"
> We do the latter, and will update the text to better emphasize this.
>
> "Figure 6 - the results here seem to indicate that the learned optimizer transfers reasonably well, achieving similar performance to first-order methods (slightly faster validation reduction). Given however that these are plots for only 10000 iterations it is still unclear if this is scalable to larger problems."
> Transfer to larger problems was not the focus of this work. We targeted stable training of task specific optimizers. In this context, 10k inner iterations is enough to achieve the best performance on these tasks (with learned optimizers). We agree that transfer to larger problems is critical for broad applicability, and are actively working on this.

---

> > ### Comment · AnonReviewer1 · 2018-11-21
> > **Weight decay**
> >
> > After reading the other reviews I want to further raise the issue that Reviewer3 raised for weight decay. I do think that it is unfair to pass to the learned optimizer as inputs the "parameter values" as indicated at the begining of the experimental section This allows them to effectively learn a weight decay update (and essentially simple prior functions over the weights) which could be the main (or even only) reason why the the proposed method perfrorms better than the baselines. Hence, I think for this experiment to be convincing you need to either exclude this term from the inputs to the MLP (and anything else that can resemble it) or alternatively include a weight decay parameter in the baseline algorithms and optimize that with respect to the final validation loss.

---

### Official Review · AnonReviewer2 · 2018-10-24
**An interesting paper too condensed and difficult to understand**

**Rating:** 5
**Confidence:** 3

**Review:**

Review:

	This paper proposes a method to learn a neural network to perform optimization. The idea is that the neural network will receive as an input several parameters, including the weights of the network to be trained, the gradient, and so on, and will output new updated weights. The neural network that is used to compute new weights can be trained through a complicated process called un-rolled optimization. The authors of the paper show two problems with this approach. Namely, the gradients tend to explode as the number of iterations increases. Truncating the gradient computation introduces some bias. To solve these problems the authors propose a variational objective that smooths the objective surface. The proposed method is evaluated on the image net dataset showing better results than first order methods optimally optimized.

Quality:

	The quality of the paper is high. It addresses an important problem of the community and it seems to give better results than first other methods.

Clarity:

	The clarity of the paper is low. It is difficult  to follow and includes many abstract concepts that the reader is not familiar with. I have had problems understanding what the truncation means. Furthermore, it is not clear at all how the validation data is used as a target in the outer-objective.  It is also unclear how the bias problem is addressed by the method proposed by the authors. They have said nothing about that, yet in the abstract they say that the proposed method alleviates the two problems detected.

Originality:

	As far as I know the idea proposed is original and very useful to alleviate, at least, one of the problems mentioned of exploding gradients.

Significance:

	It is not clear at all that the method is evaluated on unseen data when using the validation data for outer-training. This may question the significance of the results.

Pros:

	- Interesting idea.

	- Nice illustrative figures.

	- Good results.

Cons:

	- Unclear points in the paper with respect to what truncation means.

	- The validation data is used for training and there is no left-out data, which may bias the results.

	- Unclear how the authors address the bias problem in the gradients.

---

> ### Author Response · Authors · 2018-11-14
> **Thank you**
>
> Thank you for your thoughtful review. We will take these under consideration for a future submission. We have addressed your comments below.
>
> Truncation: Due to space limitations, we were unable to include a more comprehensive introduction to truncated backpropagation through time (TBTT). We would emphasize though that TBTT is a standard approach in training RNNs, and that it has an identical meaning in the case of backpropagation through many timesteps of unrolled optimization as it does in backpropagation through many timesteps of RNN dynamics. We will update the text to further emphasize this correspondence.
>
> Unseen / validation data: We agree that the distinction between validation data on train tasks and validation data on *test* tasks was unclear. We have updated the text to clarify this. To answer your question: we never see any test data before test time. This includes both the training and validation/test images. We split the Imagenet dataset by class (700 for train, 300 for test) and outer-train on the training set (using both train and validation data from those 700 classes). When evaluating our model, we use the alternate set (the remaining 300 test classes), and only use the training images for training those models.
>
> Combating biased gradients: Previous work was unable to train with longer truncations because of exploding gradients. In this work, we can simply make the truncation length longer which reduces the bias in the gradients. By unrolling longer, we drop fewer terms from the true gradient, thus lowering bias. We will make this connection clearer in the text.

---

> > ### Comment · AnonReviewer2 · 2018-11-21
> > **Response to Author Feedback**
> >
> > Thanks for the clarifications. Given them, I will slightly increase my score.

---

> > ### Comment · AnonReviewer2 · 2018-11-21
> > **Response to authors feedback**
> >
> > Although I had initially increased a bit my score, I also think that the third reviewer may have a point. Doing what the first reviewer suggest could be the way to go to guarantee fair experiments. Therefore, I have left my score unchanged.

---

### Official Review · AnonReviewer3 · 2018-11-02
**Interesting method, but oversold results**

**Rating:** 4
**Confidence:** 5

**Review:**

This paper tackles the problem of learning an optimizer, like "learning to learn by gradient descent by gradient descent" and its follow-up papers. Specifically, the authors focus on obtaining cleaner gradients from the unrolled training procedure. To do this, they use a variational optimization formulation and two different gradient estimates: one based on the reparameterization trick and one based on evolutionary strategies. The paper then uses a method from the recent RL literature to combine these two gradient estimates to obtain a variance that is upper-bounded by the minimum of the two gradients' variances.

While the method for obtaining lower-variance gradients is interesting and appears useful, the application to learn optimizers is very much oversold: the paper states that the comparison is to "well tuned hand-designed optimizers", but what that comes down to in the experiments is Adam, SGD+Momentum, and RMSProp with a very coarse grid of 11 learning rates and *no regularization* and *no learning rate schedule*. The authors' proposed optimizer is just a one-layer neural net with 32 hidden units that gets as input basically all the terms that the hand-designed optimizers compute, and it has everything it needs to simply use weight decay and learning rate schedules -- precisely what you need for the authors' contributions (speed and generalization). This is a fundamental flaw in the experimental setup (in particular the choice of baselines) and thus a clear reason for rejection.

Some details:

- While the authors' method is optimized by training 5 x 0.5 million, i.e. 2.5 million (!) full inner optimization runs of 10k steps each, the hand-designed optimizers get to try 11 values for the learning rate, which are logarithmically spaced between 10^{-4} and 10 (i.e., very coarsely, with sqrt{10} difference between successive values; even just for this fixed learning rate one would want to space factors by as little as 1.1 or so in the optimal region).

- The lack of any learning rate schedule for the baselines is highly problematic; it is common knowledge that learning rate schedules are important. This is precisely why one would want to do research on learning optimizers to set the learning rate! Of course, without learning rate schedules one will not obtain a very efficient optimizer and it is easy to show large speedups over that poor baseline (the authors' first stated contribution in the title).

- The authors' second stated contribution is that their learned optimizers generalize better than the baselines. But they pass their optimizers all information required to learn arbitrary weight decay, while the baselines are not allowed to use any weight decay. Thus, the second stated contribution in the title also does not hold up.

- There are many details in the experiments that would be hard to reproduce truthfully. Given the reproducibility crisis in machine learning, I would trust the results far more if the authors made their code available in anonymized form during the review period. If the authors did this I could also evaluate it against properly tuned baseline optimizers myself. In that case I would lean towards increasing my score since the availability of code for this line of work would be very useful for the community.

- Page 4 didn't print for me; both times I tried it came out as a blank page.

- Several issues on page 4:
 - I don't see why the unnumbered equation necessarily leads to an exponential increase; H^{(j)} can be different for each j, such that there isn't a single term being exponentiated. Or am I mistaken?
 - The problem in Figure 3a is not the problem discussed in the text
 - The global minimum of the function is not 0.5 as stated in the caption
 - It is not stated what sort of MLP there is in Figure 3d (again, code availability would fix things like this)

- Section 5 is extremely dense. This is the paper's key methodological contribution, and it is less than a page! I would suggest that the authors describe these methods in more detail (about another page) and save space elsewhere in the paper.

The paper is written well and the illustrations of the issues of TBPTT, as well as the authors' fix are convincing. It's a shame, but unfortunately, the stated contributions for the learned optimizers do not hold up.

---

> ### Author Response · Authors · 2018-11-14
> **Thank you**
>
> Thank you for your thoughtful review! We will take these under consideration for a future submission. Comments addressed below.
>
> Lack of good baseline:
> You raise a good point. We will update the paper to include a more extensive hyperparameter search for the baselines, including a denser learning rate search, and a search over regularization and learning rate decay parameters. We will also soften the claims surrounding speedups over hand designed optimizers.
>
> However, we would like to reemphasize that learned optimizers have not previously been shown to beat, or even match, standard first order optimizers on wall clock time. We would also like to reemphasize that previous approaches to meta-training learned optimizers required many tricks, and a lucky choice of random seed. A primary aim of our experiments was to provide proof that the proposed meta-training method works well and is reliable enough to train a simple mlp meta-optimizer without the complex tricks employed in other works. We believe that this work represents a significant step forward in training learned optimizers, and we are gratified to hear that you thought our analysis sections and proposed fixes were convincing.
>
> "I don't see why the unnumbered equation necessarily leads to an exponential increase; H^{(j)} can be different for each j, such that there isn't a single term being exponentiated. Or am I mistaken?"
>
> You are correct that H^{(j)} can be different for each training iteration in the general case. In a quadratic setting, H is constant, and the outer-gradient will explode if the learning rate is too large. In the non-quadratic setting (any time the Hessian is changing) it is possible for this equation to be stable depending on the sequence of Hessians encountered. Empirically, we find that the optimal meta-parameters are right on the edge of instability, so it is common to enter the unstable regime, causing outer-gradients to then grow exponentially with the number of steps.
>
> "The global minimum of the function is not 0.5 as stated in the caption":
> This is a typo and will be fixed. Should be ~3.5 (the actual global min). Thank you for spotting this.
>
> "The problem in Figure 3a is not the problem discussed in the text."
> This is discussed in the last paragraph of page 4.
>
> Reproducibility: We provide more detailed information about experiments in the appendix (page 11-12). Additionally, we are looking into options as to how to release evaluation code containing demonstrative problems and the weights of the learned optimizer.

---

> > ### Comment · AnonReviewer3 · 2018-12-05
> > **Good luck with a future submission**
> >
> > Thank you for your response, and for accepting my points of criticism & promising to fix them for future submissions. I'm looking forward to seeing this interesting work progress!
> >
> > Just a quick point about your reply concerning reproducibility: while the weights of the learned optimizer would be somewhat interesting, it would be far more useful for the community to have access to the code for training these weights. E.g., only on toy problems, so that it can be run without massive compute resources (but even if it does take a lot of compute that would still be extremely useful).
> >
> > By the way, since the paper's methodological contribution is about how the gradient signal is computed, and the definition of learned optimizers is "just" an application (and probably weaker when comparing to stronger baselines), you could consider changing the paper title to something along the lines of "Computing high quality gradient signals for unrolled computation graphs".
> >
> > Good luck with the continuation of this work!

---

> > ### Comment · AnonReviewer3 · 2018-12-08
> > **Still the same issues in the version in the MetaLearning workshop**
> >
> > I just looked into the version of this work accepted at the NeurIPS workshop on MetaLearning (http://metalearning.ml/2018/papers/metalearn2018_paper38.pdf -- warning to other reviewers: clicking this link will reveal the authors' identity), and I am disappointed to see that the issues with the experiments are not mentioned in it, even though the authors have known about them for a month.
> >
> > I am not asking for new experiments, just for explicitly stating that the authors only compare to optimizers with fixed learning rates and without any regularization. Otherwise, people will walk away from this with overblown expectations, thinking we all should use these learned optimizer (and that the only issue left is generalization to new problems, but this is simply not the case)!
> >
> > I hope that this was merely an issue of the authors forgetting to update that paper. I strongly encourage the authors to emphasize the limitations of the work and to update the camera ready copy.

---

> > > ### Author Response · Authors · 2018-12-08
> > > **Metalearning workshop**
> > >
> > > Thank you for your ongoing care as a reviewer.
> > >
> > > We have not updated the 4 page metalearning workshop paper since its submission before the review discussion — we have simply been busy.
> > >
> > > We completely agree that more extensive baselines would improve the paper. Especially, it would make our results stronger to compare against first order methods in conjunction with regularization, rather than first order methods on their own. We in fact have experiments currently running which extend the baselines in this and other fashions. We intend to include these additional baselines in any future submitted version of the paper.
> > >
> > > However, we also believe that the current baselines are accurately and clearly described, and we do not believe them to be in any sense dishonest.

---

> > > > ### Comment · AnonReviewer3 · 2018-12-23
> > > > **Issue resolved**
> > > >
> > > > The authors updated the metalearning workshop paper, resolving the issue I raised.
> > > > I'm looking forward to seeing this interesting work progress!

---

### Meta-Review · Area_Chair1 · 2018-12-13

**Confidence:** 5
**Recommendation:** Reject

**Metareview:**

The paper conveys interesting idea but need more work in terms of fair empirical study and also improvement of the writing. The AC based her summary only on the technical argumentation presented by reviewers and authors.